# Schistosomiasis outbreak during COVID-19 pandemic in Takum, Northeast Nigeria: Analysis of infection status and associated risk factors

**Francisca Olamiju[1], Obiageli J. Nebe[2], Hammed Mogaji[3]\*, Ayodele Marcus[1], Perpetua Amodu–Agbi[2], Rita O. Urude[2], Ebenezer Apake[4], Olatunwa Olamiju[1], Chimdinma Okoronkwo[1], Ijeoma Achu[1], Okezie Mpama[1]**

**1** Mission To Save The Helpless (MITOSATH), Jos, Nigeria, **2** Neglected Tropical Disease Unit, Federal Ministry of Health, Abuja, Nigeria, **3** Parasitology and Epidemiology Unit, Department of Animal and Environmental Biology, Federal University Oye-Ekiti, Oye-Ekiti, Nigeria, **4** Neglected Tropical Disease Unit, Ministry of Health, Jalingo, Taraba, Nigeria

\* mogajihammed@gmail.com

**Data Availability Statement:** We hereby inform you that there are legal restrictions on sharing the dataset, as they are owned by a third-party. Also,

## Abstract

### Background

Mass drug administration for schistosomiasis started in 2014 across Taraba State. Surprisingly in 2020, an outbreak of schistosomiasis was reported in Takum local government area. This epidemiological investigation therefore assessed the current status of infection, analyzed associated risk factors and arrested the outbreak through community sensitization activities and mass treatment of 3,580 persons with praziquantel tablets.

### Methods

Epidemiological assessment involving parasitological analysis of stool and urine samples were conducted among 432 consenting participants in five communities. Samples were processed using Kato-Katz and urine filtration techniques. Participants data on demography, water contact behavior and access to water, sanitation and hygiene facilities were obtained using standardized questionnaires. Data were analysed using SPSS 20.0 and significance level was set at 95%.

### Results

An overall prevalence of 34.7% was observed, with 150 participants infected with both species of *Schistosoma* parasite. By communities, prevalence was higher in Birama (57.7%), Barkin Lissa (50.5%) and Shibong (33.3%). By species', *S. haematobium* infection was significantly higher than *S. mansoni* (28.9% vs 9.5%), with higher proportion of younger males infected (p<0.05). The condition of WASH is deplorable. About 87% had no latrines, 67% had no access to improved source of potable water and 23.6% relied on the river as their main source of water. Infections was significantly associated with water contact behaviors

as part of the ethical guidelines, there are imposed restrictions on sharing this data except when necessary and a request for data sharing must be filed. As such, request for the dataset can be sent to Neglected Tropical Disease Unit, Federal Ministry of Health, Abuja, Nigeria (owukpa2000@gmail. com).

**Funding:** The author(s) received no specific funding for this work.

**Competing interests:** The authors have declared that no competing interests exist.

**Abbreviations:** FMoH, Federal Ministry of Health; FGS, Female Genital Schistosomiasis; GSA, Global Schistosomiasis Alliance; LGAs, Local Government Areas; MDA, Mass Drug Administration; NTD, Neglected Tropical Diseases; SPSS, Statistical Package for Social Sciences; SCI, Schistosomiasis Control Initiative; WHO, World Health Organization.

like playing in water (OR:1.50, 95% CI: 1.01–2.25) and swimming (OR:1.55, 95% CI: 1.04–2.31).

## Conclusion

It is important to reclassify the treatment needs of Takum LGA based on the findings of this study. Furthermore, efforts targeted at improving access to WASH, reducing snail population, improving health education and strengthening surveillance systems to identify schistosomiasis hotspots will be a step in the right direction

## Introduction

Schistosomiasis is an acute and chronic parasitic disease, caused by a water-borne trematode of the genus Schistosoma. Owing to the burden associated with this disease, the World Health Organization (WHO) classified it as one of the most common neglected tropical diseases (NTDs) requiring public health attention [1] Schistosomiasis is as well a focal disease [2], with a wide geographic distribution [3,4]. Currently, over 206 million people in 78 countries are affected with about 24,000 deaths and 2.5 million disability-adjusted life years (DALYs) [3]. The disease thrives in tropical and subtropical regions, especially among rural and marginalized urban populations without access to water, sanitation and hygiene (WASH) facilities [1,4–6].

It is estimated that at least 90% of those affected and requiring treatment for schistosomiasis live in Africa [4]. In this region, there are two major species of *Schistosoma*; the first is the *S. haematobium* which inhabits the vesicular and pelvic venous plexus of the bladder and causes urogenital schistosomiasis and the second is *S. mansoni* which is more often in the inferior mesenteric veins draining the large intestine and causes intestinal schistosomiasis [4,6]. In addition, the former has been reported in the Middle East and Corsica, while the latter has a wider distribution in the Middle East, the Caribbean, Brazil, Venezuela and Suriname [4]. The pathologies associated with both species vary depending on factors not limited to the severity of infection, migration of the worms through the organs and body tissues and inflammatory responses to the presence of the eggs laid by the adults [4,7,8]. Intestinal schistosomiasis can result in symptoms such as abdominal pain, diarrhea, blood in the stool, and in more severe cases, enlargement of the liver and spleen, a condition known as hepatosplenomegaly [4,9]. However, haematuria, which is classified as the passage of visible or invisible blood in urine is a common sign of urogenital schistosomiasis [4]. Other complicated pathologies may include fibrosis of the bladder and ureter, kidney damage and in more advanced cases cancer of the bladder [4]. Urogenital schistosomiasis may become more complex in females in a condition known as female genital schistosomiasis (FGS), which may present with symptoms such as genital lesions, vaginal bleeding, pain during sexual intercourse and infertility [10–12]. The pathologies are worsened among children because of the developing immunity, with already established evidence on anemia, stunting, protein-energy malnutrition, school absenteeism and reduced cognition [13–15].

Children under 15years of age remains the most vulnerable and represent the target group for most control interventions [5]. Ongoing elimination effort involves mass drug administration (MDA) of praziquantel to school-aged children between age 5 and 14 years in endemic areas following already established guidelines [5]. Since 2010, the WHO has coordinated the annual distribution of 250 million praziquantel donated by Merck and co. to several endemic

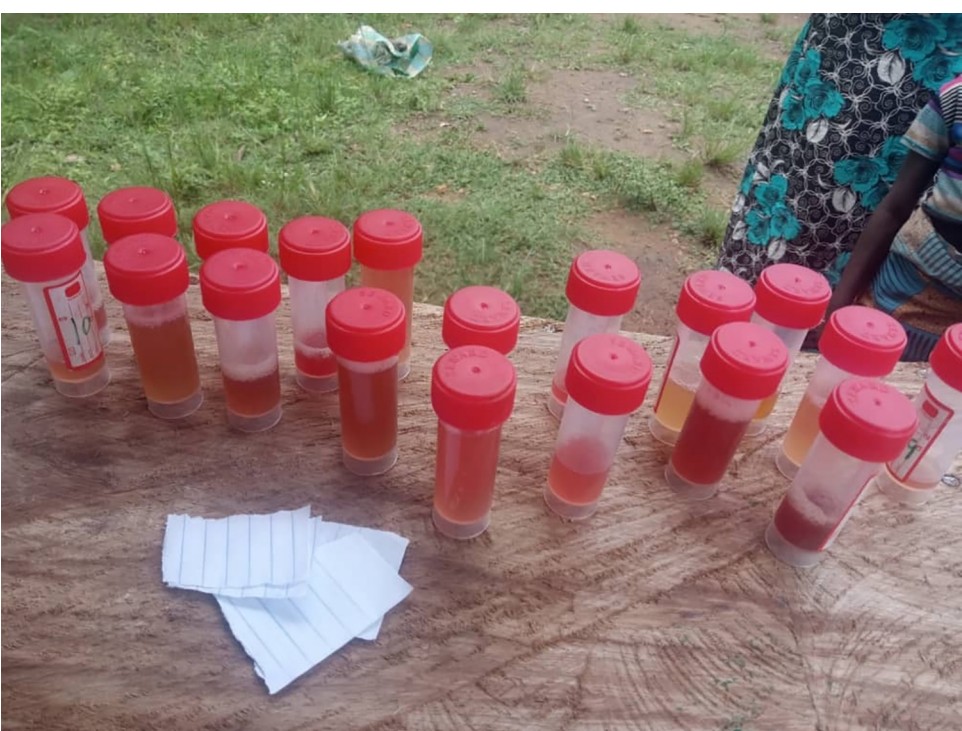

**Fig 1. Samples of bloody urine collected during study procedures.** Source: The authors took this picture on using their camera with the permission of the participants. Permission: The authors give permission to re-use this image.

countries with about 95.3 million people treated in 2019 [3]. Nigeria is one of the schistosomiasis endemic countries in Africa [1], with 36 states and 774 local government areas (LGAs). About 708 LGAs had been mapped by the Federal Ministry of Health (FMoH), with 608 of them being endemic [16]. Since 2009, treatment with praziquantel commenced in 27 states with the support of WHO, UNICEF and partner organizations such as Mission to save the helpless (MITOSATH), Sightsavers, AMEN foundation among others [16]. Taraba, was among the states in mapped for schistosomiasis in 2010 and subsequently in 2014 [16,17]. The state is located in the northeastern region of the country, and has 16 LGAs. During the mapping phase, a total of 80 schools was surveyed (5 schools per LGAs) with urine and stool examination from 3,936 school-aged children [17]. Takum was one of the LGAs mapped, with a low prevalence of 4% across the five schools examined in Sufa, Gboko, Kwambai, Takum A and Takum B communities. The LGA was then classified to be of low endemicity, and benefitted from biennial treatment strategy targeted at school-aged children since 2014 [17]. The therapeutic coverage in this LGA was optimal (>75%) in the last 5 rounds of mass drug administration (MDA) [17]. In August 2020, during the COVID-19 pandemic, an outbreak of schistosomiasis was reported in both children and adults across eleven communities in Takum (Barki Lissan, Liji, Takpa, Shibong Igbang, Lukpo, Kashimbila, Birama, Bibbi, Bawuro, Gamga and Mamga) (Fig 1). These communities were not part of the communities mapped in 2014, which calls for urgent public health action. The study was therefore conducted to (1) re-assess the prevalence of schistosomiasis in these communities; (2) document the status of water, sanitation and hygiene (WASH) resources; (3) identify risk factors promoting the transmission of schistosomiasis (4) treat the entire population and create awareness about schistosomiasis and (5) provide recommendations to improve program planning and implementation. In this paper, we, therefore, summarize the findings from the epidemiological study conducted and

the programmatic actions implemented in line with the global target of eliminating schistosomiasis.

## Methodology

### Ethical statement and considerations

Ethical clearance for this study was obtained from the Taraba State Ministry of Health ethics review board. A pre-survey contact/advocacy meeting was made to each selected study community to obtain consent from community leaders and other major stakeholders after explaining the objectives of the research to them. This was followed by community mobilization and sensitization using town announcers to communicate the objectives of our visit to community members. Sensitization was done in all religious and public places like schools and market squares to promote participation. Community members willing to participate in the study completed written consent forms on the day of sample collection. Assent forms was completed in cases where the willing member is below 16 years of age. In this case, parents or any legal guardian were asked to accompany minors under age 16 to the sample collection site, to provide additional consents. The method of consent assertion was through thumbprint on already printed informed consent forms (ICFs).

### Study area

This study was carried out in five communities located in Takum LGA, Taraba state, Northeastern, Nigeria. Takum is one of the 16 LGAs in Taraba state, with an approximate land area of 2,503 km$^2$ (Fig 2) [17]. The climate of the area is tropical with vegetation characterized by a typical Guinea savannah interspersed with gallery forest. The annual rainfall ranges between 1,200mm and 2,000mm annually, while the average temperature is between 28 and 32˚C

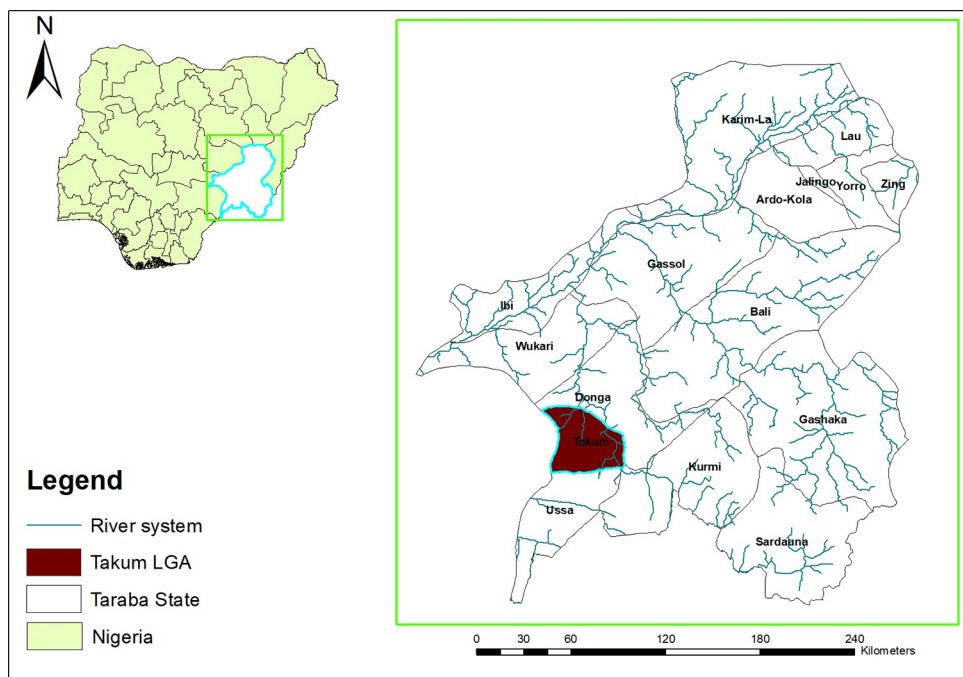

**Fig 2. Map of Taraba State showing the study LGA.** Source: The authors using their primary data in ArcGIS software created this map. Permission: The authors give permission to re-use this map.

reaching a peak at 37˚C in March and April. In addition, the area has several ponds, streams and rivers, which provides conducive environment for farming and fishing occupation, as well as sites for other recreational activities such as bathing, swimming, and washing of clothes [17].

## Study design and selection of communities

This study employed a cross-sectional sampling design involving questionnaire administration and sample collection in five communities out of the eleven communities that reported schistosomiasis outbreak in the LGA. The five study communities (Barkin lissa, Birama, Gamga, Shibong and Takpa) were randomly selected using the paper ballot approach out of the eleven communities. After selection, the severity of the outbreak reported to the district health officer was re-examined, and compared among selected communities and those that were not selected. Replacements were done where necessary to ensure a balance of priority. Preliminary advocacy visits were made to the NTD control unit closer to the selected communities (i.e., ward level), prior to the epidemiological investigation. The study was conducted in September, 2020, and involved 4 distinct phases; (1) advocacy and sensitization; (2) questionnaire administration; (3) sample collection and laboratory examination and (4) treatment of all persons.

## Sample size determination and selection of study participants

As an initial step, the total population of school-aged children and adults in the communities were extracted from the 2020 village census register obtained from the NTDs control department in the LGA. A total of 6,012 persons comprising 2531 children and 3481 adults were enumerated. As a further step, a sample-size was determined using the formula; $n_s = \frac{n}{1+(n/N)}$ and $n = \frac{z^2 p(1-p)}{d^2}$, as described by Lwanga et al [18], where $n_s$ is the required sample size and N is the target population. We assumed a prevalence (p) of 50% since there are no previous baseline data on schistosomiasis in the five communities, a relative precision (d) of 5% and a confidence level of 95% which corresponds to a z score of 1.96. The minimum sample size determined, therefore, was 362 i.e., an average of 72 persons per community. However, the recruitment of participants extended beyond the estimated sample size, considering the aim of identifying factors associated with the outbreak.

For the selection of study participants, we employed a total sampling methodology, following the method previously described by [19]. Community sensitization and advocacy visits were made to the community leaders and other stakeholders. This was followed by mobilization of eligible community members to participate in the study using town announcers Only residents of the community, who are above age of 5 years, can provide consent or assent, and has lived within the community in the last 3 months were recruited into the study. Enrollment of participants and collection of samples took place at a central point in the community, provided by the community leader. This point has a secluded space for administering study questionnaires and sorting of samples before transporting them to the laboratory. The number of consenting participants varied across the communities, hence giving an unequal number of persons recruited.

## Questionnaire administration

A simple standardized questionnaire was used to collect participants' details (S1 Appendix). The questionnaire tool which has four different sections; demographic, WASH, water contact practice and laboratory results, was first designed in English language (S1 Appendix), translated to Hausa language (S2 Appendix), piloted and designed as an electronic data from. Prior

to administration of questionnaire, recruited participants who had completed informed consent or assent forms were assigned unique identification number. The demographic section of the questionnaire captured the name, sex, age and unique identification number of the participant. Participants unique identification number was used to allocate a pre-labeled sterile stool and urine specimen bottle. Furthermore, the WASH section captured information about participants' access to water, sanitation and hygiene facilities. The water contact practice section was used to document the range of water contact activities the participants performed in the last three months, while the laboratory section was used to document the findings from the laboratory assessment of the urine and stool specimen. Before data collection, research assistants were trained on how to capture data electronically using Kobo collect tool and LINKS system on smartphones. All data were collected electronically and transferred to a remote backup server immediately after each interview. All interviews were conducted in Hausa language and held in confidence in a private space, except when the interviewee is a minor and needs the assistance of a legal guardian or a parent.

## Collection of stool and urine samples

Participants were provided with two sterile specimens bottle, pre-labeled with their unique identification number, an applicator stick, a plain sheet of paper and a tissue paper to clean their anus. Participants were instructed to defecate on the plain sheet of paper and use the applicator stick to transfer a fresh portion into the first bottle. Furthermore, they were instructed to provide approximately 10ml of urine in the second bottle. Samples bottled were retrieved within 1 hour of distribution. All eligible and non-eligible participants were treated with 400mg/kg of praziquantel as an immediate benefit of the research investigation. In addition to this, incentives such as bar soaps and bisquits were given to reinforce positive hygiene behaviors and stimulate community integration during a tensed situation which the pandemic has presented. These incentives were provided in a non-coercive manner, ensuring they do not influence the decision of the community members to participate or decline participation in the research.

## Parasitological assessment of stool and urine samples

All collected stool and urine samples were sorted and transported for processing within 2hours of collection to the Parasitology laboratory located in Takum General Hospital. The urine filtration method was employed to identify *S. haematobium* eggs. In brief, 10ml of urine sample was vigorously shaken and passed through a Nytrel filter with a 40 μm mesh size. The filter was then placed on a clean microscopic slide and viewed under the microscope using the x10 and x40 objective lens in search of an egg with a characteristic terminal spine. For each slide, the fields were re-examined and eggs were re-counted by another microscopist for quality assurance. Similarly, stool specimens were processed using the Kato-Katz technique. Two thick smears were prepared from a single stool sample and allowed to clear for 30 minutes before microscopic examination for *S. mansoni*. The fields were also re-examined and counter-check by another microscopist. For both urine and stool specimens, a participant is considered infected, if there is an egg count recorded on both sheets of the two microscopists who examined the smears.

## Treatment of all consenting persons and sensitization about schistosomiasis

Following field procedures, the NTD unit at the sub-district level performed a door-to-door administration of praziquantel (400mg/kg) to all persons in the community. During their

visits, they sensitized the household members about schistosomiasis and the need to avoid contact with the river. They also emphasized prompt reporting of symptoms such as bloody urine to the nearest health center. The field team was supervised by a team comprising the NTD coordinators from the FMoH, the state and the LGA.

**Data management and analysis.** Data obtained were downloaded from the remote server by the biostatistician, and imported into Microsoft Excel for sorting before analysis in SPSS 20.0 software. Data on socio-demographic characteristics and water contact behavior were considered as independent variable, while prevalence of infection was considered as dependent variable. Data obtained were first subjected to descriptive statistics including frequencies and cross-tabulations, then followed by Pearson chi-square statistics to test for associations between the variables. Variables that were associated with infection were considered significant only when $P < 0.05$. Subsequently, variables were also subjected to univariate analyses i.e., logistic regression, to estimate the magnitude of association between infection data and other variables. Potential risk factors were entered into the model as covariates using bidirectional stepwise entry method. Reference category was formulated for categorical variables before analysis and observations with missing values for any variable were excluded from the analysis. Predictive index in the model is represented as Exp(B) which is the odds-ratio (OR). A 95% confidence interval (CI) was calculated for the OR, and values were considered statistically significant when the CI does not include 1 and the $P < 0.05$.

## Result

### Demographic characteristics of study participants

A total of 432 community residents from five communities; Barkin lissa (97, 22.5%), Birama (71, 16.4%), Gamga (76, 17.6%), Shibong (96, 22.2%) and Takpa (92, 21.3%) were enrolled into this study. The majority of the participants were males (218, 50.5%), compared to females (214, 49.5%), and there was a significant difference in the gender distribution across the communities (p = 0.00). By age category, the majority of the participants were between age 5 and 10 (152, 35.2%), followed by those above 21 years (130, 30.1%), 11–16 years (112, 25.9%) and 17–20 years (38, 8.8%). There were also significant differences between the age category of participants across the study communities (p = 0.02) (Table 1).

### Prevalence of schistosomiasis among the study participants

Of the 432 participants examined, a total of 150 (34.7%) were infected with both species of Schistosoma parasite; 125 (28.9%) for *S. haematobium*, and 41 (9.5%) for *S. mansoni*.

**Table 1. Demographic characteristics of the study population.**

| | Communities | | | | | | |
|---|---|---|---|---|---|---|---|
| | Barkin Lissa (n = 97) | Birama (n = 71) | Gamga (n = 76) | Shibong (n = 96) | Takpa n = 92) | Total (n = 432) | $X^2$, df, pvalue |
| **Sex** | | | | | | | |
| Female | 36(37.1) | 43(60.6) | 44(57.9) | 39(40.6) | 52(56.5) | 214(49.5) | 16.412, 4, 0.00 |
| Male | 61(62.9) | 28(39.4) | 32(42.1) | 57(59.4) | 40(43.5) | 218(50.5) | |
| **Age group in years** | | | | | | | |
| 5–10 | 31(32.0) | 32(45.1) | 27(35.5) | 28(29.2) | 34(37.0) | 152(35.2) | 23.595,12, 0.02 |
| 11–16 | 26(26.8) | 17(23.9) | 22(28.9) | 32(33.3) | 15(16.3) | 112(25.9) | |
| 17–20 | 16(16.5) | 1(1.4) | 4(5.3) | 8(8.3) | 9(9.8) | 38(8.8) | |
| >21 | 24(24.7) | 21(29.6) | 23(30.3) | 28(29.2) | 34(37.0) | 130(30.1) | |

**Table 2. Prevalence of schistosomiasis among the study participants.**

| Communities | NE | *S. haematobium* | | | *S. mansoni* | | *S haematobium +S. mansoni* | |
|---|---|---|---|---|---|---|---|---|
| | | NI | 95% CI | | NI | 95% CI | NI | 95% CI |
| Barki Lissa | 97 | 49 | 50.5 (40.5–60.5)[c] | | 6 | 6.2 (1.4–10.9) [a] | 49(50.5) | 50.5 (40.6–60.5) [c] |
| Birama | 71 | 18 | 25.4 (15.2–35.4)[b] | | 33 | 46.5 (34.9–58.1) [b] | 41(57.7) | 57.7 (46.3–69.2) [c] |
| Gamga | 76 | 12 | 15.8 (7.6–23.9) [b] | | 0 | - | 12(15.8) | 15.8 (7.6–23.9) [b] |
| Shibong | 96 | 30 | 31.3 (21.9–40.5) [b] | | 2 | 2.1(-0.78–4.94) [a] | 32(33.3) | 33.3 (23.9–42.8) [b] |
| Takpa | 92 | 16 | 17.4 (9.6–25.1) [b] | | 0 | - | 16(17.4) | 17.4 (9.6–25.1) [b] |
| Total | 432 | 125 | 28.9 (24.7–33.2) [b] | | 41 | 9.5 (6.73–12.3)[a] | 150(34.7) | 34.7 (30.3–39.2) [b] |

**NE:** Number Examined; **NI:** Number Infected; **CI:** Confidence Interval.

Categories of Endemicity

[a]Low endemicity when prevalence is between 1–9.9%.

[b]Moderate endemicity when prevalence is between 10–49.9%.

[c]High endemicity when prevalence is above 50%.

Prevalence level varies across the communities, with the highest recorded in Birama (57.7%), followed by Barkin Lissa (50.5%), Shibong (33.3%), Takpa (17.4%) and Gamga (15.8%). (Table 2). Prevalence was higher among males and children below age 16 (Figs 3 and 4). By species' prevalence, *S. haematobium* infection was significantly higher among males (P<0.05), but there was no significant difference in the proportion of males or females infected with *S. mansoni* (p>0.05) (Fig 3).

## Access to water, sanitation and hygiene (WASH) facilities and prevalence of schistosomiasis

Table 3 shows the status of water supply, sanitation and hygiene (WASH) facilities. The majority of the study participants (288, 66.7%) had no regular source of potable water supply, while a high percentage of them relied on the river as their main source of water supply (102, 23.6%). Only 6.5% of the participants had access to the handpump borehole. Furthermore, the majority of the participants had no latrines (375, 86.8%) and over 40% of them had no handwashing facilities. Of all the WASH variables examined, only access to river was significantly associated with reduced odds of infection (OR:0.27; 95% CI: 0.1–0.66).

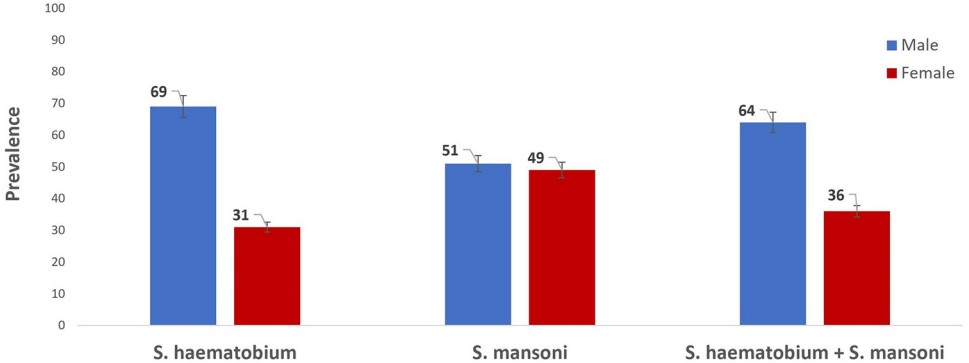

**Fig 3. Prevalence of schistosomiasis by sex among the study participants.** Source: The authors using their primary data to create this chart in Microsoft Excel software. Permission: The authors give permission to re-use this map.

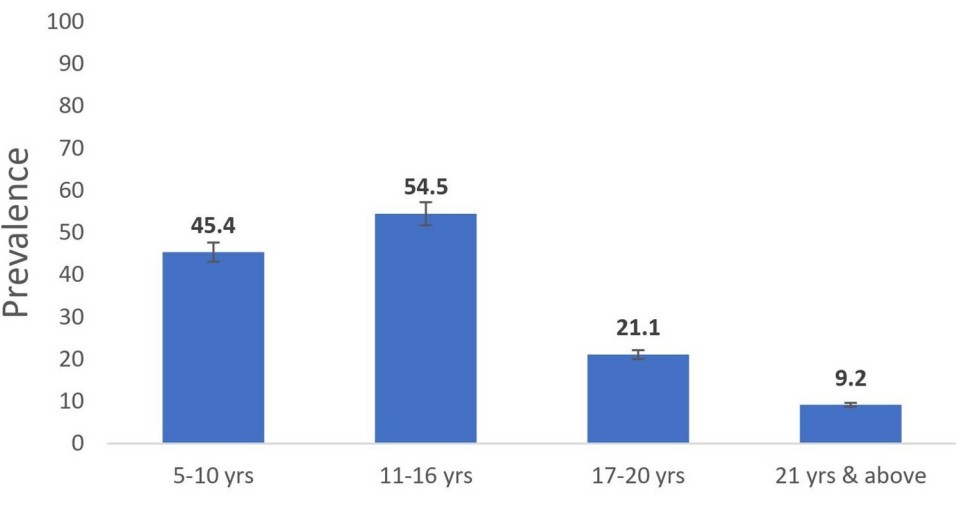

**Fig 4. Prevalence of schistosomiasis by age among the study participants.** Source: The authors using their primary data to create this chart in Microsoft Excel software. Permission: The authors give permission to re-use this map.

## Water contact behavior among the study participants

Out of the six water contact practices investigated, fishing (43, 10%) was the least common practice in the study areas. However, the majority of the participants engage in activities such as bathing (419, 97%), washing of clothes (357, 82.6%), fetching water (358, 82.9%), playing in river (193, 44.7%) and swimming (214, 49.5%). Infections were significantly associated with

**Table 3. Access to water, sanitation and hygiene (WASH) facilities and prevalence of schistosomiasis.**

| Characteristic | Frequency (%) | Positives (%) | Negatives (%) | $X^2$, df, pvalue | OR (95% CI) | p-value |
|---|---|---|---|---|---|---|
| | N = 432 | N = 150 (34.7) | N = 282 (65.3) | | | |
| **Water supply facilities** | | | | | | |
| Handpump/Borehole | 28(6.5) | 12(42.9) | 16(57.1) | 21.235, 6, 0.00 | 1 | - |
| Unprotected Dug well | 1(0.2) | 1(100.00) | 0(0.0) | | 3.96(0.14–105.6) | 0.41 |
| Protected Dug well | 6(1.4) | 2(33.3) | 4(66.7) | | 0.67 (0.10–4.26) | 0.67 |
| Surface Water (River) | 102(23.6) | 17(16.7) | 85(83.3) | | 0.27(0.1–0.66) | 0.00* |
| Rainwater collection | 3(0.7) | 1(33.3) | 2(66.7) | | 0.67(0.05–8.24) | 0.75 |
| Sachet/Pure water | 4(0.9) | 2(50.0) | 2(50.0) | | 1.33(0.16–10.86) | 0.78 |
| None | 288(66.7) | 115(39.9) | 173(60.1) | | 0.88(0.40–1.94) | 0.76 |
| **Toilet facilities** | | | | | | |
| Flush toilet | 10(2.3) | 6 (60.0) | 4 (40) | 13.59, 4, 0.01 | 1 | - |
| Pit latrine without slab | 12(2.8) | 8(66.7) | 4 (33.3) | | 1.33(0.23–7.62) | 0.32 |
| Pit latrine with slab | 22(5.1) | 12(54.5) | 10 (45.5) | | 0.80 (0.17–3.65) | 0.77 |
| VI-Pit latrine | 13(3.0) | 5 (38.5) | 8 (61.5) | | 0.42 (0.08–2.25) | 0.31 |
| None | 375(86.8) | 119(31.7) | 256 (68.3) | | 0.31(0.09–1.12) | 0.07 |
| **Handwashing facilities** | | | | | | |
| Water and Soap | 223(51.6) | 75 (33.6) | 148 (66.4) | 0.24, 1, 0.62 | 1 | - |
| None | 209(48.4) | 75 (35.9) | 134 (64.1) | | 1.1(0.74–1.64) | 0.62 |

$X^2$: Chi-square value; df: degree of freedom; OR: Odd ratio; CI: Confidence interval; * significant difference exist at 95%; VI-Pit Latrine: Ventilated Improved Pit Latrine.

**Table 4. Water contact activities and prevalence of schistosomiasis.**

| Water Contact Activities | Frequency (%) N = 432 | Positives (%) N = 150 (34.7) | Negatives (%) N = 282 (65.3) | $X^2$, df, pvalue | OR (95%CI) | pvalue |
|---|---|---|---|---|---|---|
| **Bathing** | | | | | | |
| No | 13(3.0) | 2 (0.5) | 11 (2.5) | | 1 | |
| Yes | 419(97.0) | 148 (34.4) | 271 (62.7) | 2.21, 1, 0.14 | 3.00(0.66–13.7) | 0.15 |
| **Washing** | | | | | | |
| No | 75(17.4) | 27 (6.3) | 48 (11.1) | | 1 | |
| Yes | 357(|82.6) | 123 (28.5) | 234 (54.2) | 0.06,1, 0.79 | 0.93(0.56–1.57) | 0.79 |
| **Fishing** | | | | | | |
| No | 389(90.0) | 132 (30.6) | 257 (59.5) | | 1 | |
| Yes | 43(10.0) | 18 (4.2) | 25 (5.8) | 1.074,1,0.30 | 1.40(0.73–2.66) | 0.85 |
| **Fetching Water** | | | | | | |
| No | 74 (17.1) | 27 (6.3) | 47 ((10.9) | | 1 | |
| Yes | 358(82.9) | 123 (28.5) | 235 (54.4) | 0.12, 1, 0.73 | 0.91(0.54–1.53) | 0.73 |
| **Playing** | | | | | | |
| No | 239(55.3) | 73 (16.9) | 166 (38.4) | | 1 | |
| Yes | 193(44.7) | 77 (17.8) | 116 (26.9) | 4.12, 1, 0.04 | 1.50(1.01–2.25) | 0.04* |
| **Swimming** | | | | | | |
| No | 218(50.5) | 65 (15.0) | 153 (35.4) | | 1 | |
| Yes | 214(49.5) | 85 (19.7) | 129 (29.9) | 4.67,1,0.03 | 1.55(1.04–2.31) | 0.03* |

$X^2$: Chi-square value; df: degree of freedom; OR: Odd ratio; CI: Confidence interval; * significant difference at 95%.

playing and swimming activities with increased odds of 1.50 (95% CI: 1.01–2.25) and 1.55 (95% CI: 1.04–2.31), respectively. (Table 4).

## Treatment data

A total 3,580 person were treated across the study communities. More persons were treated in Shibong (n = 1,057), followed by Birama (n = 1,044), Barki Lisa (n = 634), Gamga (n = 632) and Takpa (n = 213). Furthermore, treated males (n = 1912) were more than treated females (n = 1668), and treated persons above aged 15 (n = 2,436) were more than school-aged children between age 5 and 14 (n = 1124) (Table 5).

## Discussion

Since 2014, Takum LGA has benefitted from three biennial rounds of MDA targeted at school-aged children. In these years, the therapeutic coverage was optimal, surpassing the 75%

**Table 5. Praziquantel treatment data across the study communities.**

| | Treatment data | | | | | | | | |
|---|---|---|---|---|---|---|---|---|---|
| | 5–14 years | | | 15 years and above | | | Total treated | | |
| Communities | Male | Female | Total | Male | Female | Total | Male | Female | Total |
| Barki Lissa | 134 | 129 | 263 | 188 | 183 | 371 | 322 | 312 | 634 |
| Birama | 143 | 122 | 265 | 494 | 285 | 779 | 637 | 407 | 1,044 |
| Gamga | 103 | 68 | 171 | 231 | 230 | 461 | 334 | 298 | 632 |
| Shibong | 186 | 145 | 311 | 328 | 398 | 726 | 514 | 543 | 1,057 |
| Takpa | 56 | 58 | 114 | 49 | 50 | 99 | 105 | 108 | 213 |
| Total | 622 | 522 | 1124 | 1290 | 1146 | 2436 | 1912 | 1668 | 3580 |

national targets [17], as such, the outbreak of schistosomiasis in this area was unexpected. Furthermore, with the advent of COVID-19, MDA program was paused in 2020 across endemic countries, owing to the fact that mass gatherings during trainings and administration of medicines may increase the transmissibility of the virus [20]. The shifts in policies to non-pharmaceutical interventions including closure of schools and restrictions placed on public gatherings and movements therefore impacted on the response time of the epidemiology team during this outbreak. Nevertheless, the team arrested the outbreak through mass treatment of all eligible persons above age 5 in concordance with the standard operating guidelines stipulated for resuming MDA amid COVID-19 pandemic [21]. It is therefore necessary to present the learnings from the epidemiological analysis of the outbreak, more importantly, the current status of infection, associated risk factors and recommendations to forestall future occurrence.

The prevalence reported in this study corroborates with the schistosomiasis outbreak, with two of the communities having an overall prevalence above 50%, another had a prevalence above 30% and two communities had their prevalence between 16 and 17%. The moderate prevalence (<50%) recorded in the other three communities could be attributed to the fact that targeted administration of Praziquantel was carried out before the arrival of the epidemiological team. Also, on an aggregated basis, the pattern of infection across the communities might have been masked, since the prevalence across these five communities is 34.7%. This aggregation could misinform program actions targeted at eliminating the disease [19]. Until now, Takum was classified to be a low endemic LGA, and had been receiving biennial treatment [17]. This outbreak and our prevalence reports, therefore, highlight the need to re-classify the LGA for annual treatment, and also support the ongoing discussion on precision mapping and disaggregation of data during planning and implementation of MDA [2]. This becomes very important considering the focality of schistosomiasis, and the complex life cycle involving a mixture of human behavior and availability of snail intermediate host in conducive water bodies.

WASH has been advocated severally as a complementary tool to ongoing MDA program focused on schistosomiasis [22–24]. Surprisingly, the odds of infection reduced among those who regard the river as their source of drinking water. The collection of water for drinking has been reported as a relatively less important pathway of infection because it involves immersion of small areas of the body and for relatively short durations unlike other activities like bathing, swimming or playing [23,25]. To support this submission, our results show that other water contact activities such as playing and swimming, which would require more contact time with the river were significantly associated with increased odds of infection, with those who visit the river to swim or play been twice more exposed than those who do not. This finding conforms with earlier reports of [23].

Swimming and playing are risk factors that are common among male young school-aged children [26,27]. Our findings also support this as the majority of those infected participants in our study were young male children below age 15. It is possible that the primary source of this outbreak might be from a segment of these young population who got in contact with the water body via swimming or playing, urinated in the process around the peak periods, thus supporting the transmission of schistosomiasis. This thought is in line with a similar outbreak reported in Zimbabwe [28]. This segment of the population might have been heavily infected and under-treated because of the previous misclassification of the LGA on treatment basis. On the other hand, the closure of schools during the pandemic era supports clustering and more contact hours between young school-aged children at river sites, from different communities and could also be another pathway of contamination of river sites [29].

Notwithstanding, the epidemiological risk analysis has raised the following substantial concerns that could have supported the outbreak; (i) lack of baseline mapping in the study

communities which calls for more refined approaches such as precision mapping, (2) misclassification of the LGA based on treatment needs which resulted in undertreatment (3) predominant risky behavior of swimming and playing among the young children which might have been compounded by the lockdown imposed from the pandemic, and (4) availability of a pool of viable intermediate snail host at the river sites. These concerns, therefore, reflect gaps that need to be addressed in line with the goal of eliminating schistosomiasis by 2030.

It is therefore imperative to consider; (1) investments in effort targeted at reclassifying the LGA as highly endemic, and adjusting the MDA thresholds from the biennial cycle to an annual cycle (2) strengthening surveillance system to identify hot-spots such as areas with high reportage of hematuria, (3) investments in the epidemiological mapping of infections when resources allow, (4) continuous sensitization of young children, most especially as schools have resumed on the dangers of excessive recreational activities at the river site is important, and (5) investments in efforts targeted at reducing the snail population in the river body associated with these communities.

## Conclusion

Until now, Takum was classified to be a low endemic LGA and had been receiving biennial treatment. This outbreak and our prevalence reports highlight the need to re-classify the LGA as highly endemic, and adjust the MDA thresholds from the biennial cycle to an annual cycle. In addition, our findings support the ongoing discussion on precision mapping and disaggregation of data during planning and implementation. Swimming and playing in rivers were the most potent risk factor supporting the transmission of schistosomiasis. Strengthening available surveillance systems to identify hotspots and investments in efforts targeted at improving health education of children and reducing snail population will be a step in the right direction.

## Limitation of the study

Although participation was voluntary, some participants might be afraid to join the study because of their perception of the pandemic. As such, we cannot ignore the impact, the COVID-19 pandemic had on recruitment of participants.

## Supporting information

**S1 Appendix. Questionnaire for the study in English Language.**
(DOCX)

**S2 Appendix. Questionnaire for the study in Hausa Language.**
(DOCX)

## Acknowledgments

We are grateful to the community leaders across the study areas, and all the health workers who gave sacrificed their time during the outbreak despite the pandemic situation.

## Author Contributions

**Conceptualization:** Francisca Olamiju, Obiageli J. Nebe, Ebenezer Apake, Olatunwa Olamiju, Ijeoma Achu.

**Data curation:** Hammed Mogaji, Perpetua Amodu–Agbi, Rita O. Urude, Chimdinma Okoronkwo.

**Formal analysis:** Hammed Mogaji.

**Funding acquisition:** Francisca Olamiju, Olatunwa Olamiju.

**Investigation:** Francisca Olamiju, Obiageli J. Nebe, Hammed Mogaji, Ayodele Marcus, Perpetua Amodu–Agbi, Rita O. Urude, Ebenezer Apake, Chimdinma Okoronkwo, Ijeoma Achu, Okezie Mpama.

**Methodology:** Francisca Olamiju, Obiageli J. Nebe, Ayodele Marcus, Perpetua Amodu–Agbi, Rita O. Urude, Ebenezer Apake, Olatunwa Olamiju, Chimdinma Okoronkwo, Ijeoma Achu, Okezie Mpama.

**Project administration:** Francisca Olamiju, Ebenezer Apake, Olatunwa Olamiju, Ijeoma Achu.

**Resources:** Francisca Olamiju, Olatunwa Olamiju.

**Software:** Hammed Mogaji.

**Supervision:** Francisca Olamiju, Obiageli J. Nebe, Ayodele Marcus, Perpetua Amodu–Agbi, Rita O. Urude, Olatunwa Olamiju, Ijeoma Achu.

**Validation:** Ayodele Marcus, Perpetua Amodu–Agbi, Rita O. Urude, Olatunwa Olamiju, Ijeoma Achu, Okezie Mpama.

**Visualization:** Hammed Mogaji.

**Writing – original draft:** Hammed Mogaji.

**Writing – review & editing:** Francisca Olamiju, Obiageli J. Nebe, Hammed Mogaji, Ayodele Marcus, Perpetua Amodu–Agbi, Rita O. Urude, Ebenezer Apake, Olatunwa Olamiju, Chimdinma Okoronkwo, Ijeoma Achu, Okezie Mpama.

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
