## [Decision Letter · Decision Letter 0]

13 Oct 2021

PONE-D-21-30831Schistosomiasis outbreak during COVID-19 pandemic in Takum, Northwest Nigeria: Analysis of Infection status and associated risk factorsPLOS ONE

Dear Dr. Mogaji,

Thank you for submitting your manuscript to PLOS ONE. After careful consideration, we feel that it has merit but does not fully meet PLOS ONE’s publication criteria as it currently stands. Therefore, we invite you to submit a revised version of the manuscript that addresses the points raised during the review process.

We look forward to receiving your revised manuscript.

Kind regards,

Khin Thet Wai, MBBS, MPH, MA

Academic Editor

PLOS ONE

Journal Requirements:

5. We note that Figure 2 in your submission contain [map/satellite] images which may be copyrighted. All PLOS content is published under the Creative Commons Attribution License (CC BY 4.0), which means that the manuscript, images, and Supporting Information files will be freely available online, and any third party is permitted to access, download, copy, distribute, and use these materials in any way, even commercially, with proper attribution. For these reasons, we cannot publish previously copyrighted maps or satellite images created using proprietary data, such as Google software (Google Maps, Street View, and Earth). For more information, see our copyright guidelines: http://journals.plos.org/plosone/s/licenses-and-copyright.

Additional Editor Comments:

This is the study beneficial for public health control programs in the context of the COVID 19 pandemic.

In addition to the comments of two reviewers, authors should add the following to improve scientific integrity:

1. The reported number of cases of COVID-19 in the study area which impeded the schistosomiasis control program activities.

2. The total population of school-aged children and the adult population in the study area

3. Specific computation of sample size and comprehensive sampling procedure among the target population and to include the sampling strategy related to mitigation of selection bias.

4. Comprehensive description of the simple logistic regression procedure in the data management and data analysis section.

5. To add 95% confidence intervals for prevalence ratios in Table 2.

6. Both unadjusted and adjusted odds should be included in Tables 3 and 4.

Reviewers' comments:

Reviewer's Responses to Questions

**Comments to the Author**

1. Is the manuscript technically sound, and do the data support the conclusions?

Reviewer #1: Yes

Reviewer #2: Partly

2. Has the statistical analysis been performed appropriately and rigorously? 

Reviewer #1: Yes

Reviewer #2: No

3. Have the authors made all data underlying the findings in their manuscript fully available?

Reviewer #1: Yes

Reviewer #2: Yes

4. Is the manuscript presented in an intelligible fashion and written in standard English?

Reviewer #1: Yes

Reviewer #2: No

5. Review Comments to the Author

Reviewer #1: The manuscript is well written. The edits are very few such as italicizing Schistosoma and just minimal grammatical lapses and better sentence construction. My only major question is the sample size, how it was computed and what proportion of the computed sample size is represented by the number of actual respondents.

Reviewer #2: I would advice you look at the gap and discuss your results with the information there-in.

The paper is addressing a key issue in public health i.e An outbreak of schistosomiasis in Takum LGA in Northeastern Nigeria. However, the paper has some shortfalls and I suggest you consider the following:

1. The methodology needs some clarification, administration of drugs to people was larger than the sample surveyed. The use of error bar seems misleading and needed to be re-address

2. Though the paper was well presented, however, some aspect needed for follow a chronological order so as to have free flow when reading the paper if accepted for publication.

3. The gap which was identified needed to be filled through right discussion on good implementation research

4. The discussion needs to be aligned to the key results of the study and avoid reference to stuff that was note presented in the results.

5. The present manuscript requires editing for grammar, spelling errors and further concision.

6. PLOS authors have the option to publish the peer review history of their article (what does this mean?). If published, this will include your full peer review and any attached files.

Reviewer #1: No

Reviewer #2: **Yes: **Opeyemi G. Oso

---

## [Author Response · Author response to Decision Letter 0]

25 Oct 2021

Editor Comments:

This is the study beneficial for public health control programs in the context of the COVID 19 pandemic.

In addition to the comments of two reviewers, authors should add the following to improve scientific integrity:

C1. The reported number of cases of COVID-19 in the study area which impeded the schistosomiasis control program activities.

R1: Thank you very much for the comments. The data on the number of cases of COVID-19 in Nigeria is reported at the district level (i.e., LGA). The total number of cases in Takum LGA as at the time of the outbreak (September) was 5, and this was recorded in June, as such there were no new cases since June till September. However, what impeded the schistosomiasis control activities is not the number of cases, but the concerns that such program which are characterized by mass gatherings, door to door movements, and administration of medicines to residents, has a strong potential to contribute to COVID-19 transmission. As such, the WHO mandated that all mass treatment programs should be ceased until recently.

We have now modified the beginning paragraphs of our discussion line to address this issue. We added 2 new references to support this line of discussion. 

20 Warren, L. Community directed treatment for Neglected Tropical Diseases in a post COVID-19 lockdown world. 2020. Retrieved from https://end.org/community-directed-treatment-for-neglected-tropical-diseases-in-a-post-COVID-19-lockdown-world/

21 Molyneux D, Bush S, Bannerman R, Downs P, Shu'aibu J, et al. Neglected tropical diseases activities in Africa in the COVID-19 era: the need for a "hybrid" approach in COVID-endemic times. Infect Dis Poverty. 2021;10(1):1.

We appreciate you for raising this very important issue. Please see the modified lines in the discussion below.

Discussion 

Since 2014, Takum LGA has benefitted from three biennial rounds of MDA targeted at school-aged children. In these years, the therapeutic coverage was optimal, surpassing the 75% national targets [17], as such, the outbreak of schistosomiasis in this area was unexpected. Furthermore, with the advent of COVID-19, MDA program was paused in 2020 across endemic countries, owing to the fact that mass gatherings during trainings and administration of medicines may increase the transmissibility of the virus [20]. The shifts in policies to non-pharmaceutical interventions including closure of schools and restrictions placed on public gatherings and movements therefore impacted on the response time of the epidemiology team during this outbreak. Nevertheless, the team arrested the outbreak through mass treatment of all eligible persons above age 5 in concordance with the standard operating guidelines stipulated for resuming MDA amid COVID-19 pandemic [21] It is therefore necessary to present the learnings from the epidemiological analysis of the outbreak, more importantly, the current status of infection, associated risk factors and recommendations to forestall future occurrence.

C2: The total population of school-aged children and the adult population in the study area

R2: Thank you very much for this comment, we have now included the population of the school-aged children and the adult population. We have also described extensively how we used these figures to estimate the sample size.

Please see below

Sample size determination and selection of study participants

As an initial step, the total population of school-aged children and adults in the communities were extracted from the 2020 village census register obtained from the NTDs control department in the LGA. A total of 6,012 persons comprising 2531 children and 3481 adults were enumerated. As a further step, a sample-size was determined using the formula; n_s=n/(1+(n/N) ) and n= (z^2 p(1-p))/d^2 , as described by Lwanga et al [18], where ns is the required sample size and N is the target population. We assumed a prevalence (p) of 50% since there are no previous baseline data on schistosomiasis in the five communities, a relative precision (d) of 5% and a confidence level of 95% which corresponds to a z score of 1.96. The minimum sample size determined, therefore, was 362 i.e., an average of 72 persons per community. However, the recruitment of participants extended beyond the estimated sample size, considering the aim of identifying factors associated with the outbreak.

C3: Specific computation of sample size and comprehensive sampling procedure among the target population and to include the sampling strategy related to mitigation of selection bias.

R3: Thank you very much for your very valuable comment, which has improved the quality of our manuscript. We have now provided very explicit details on how the sample-size were estimated and how the participants were recruited.

Please find below

Sample size determination and selection of study participants

As an initial step, the total population of school-aged children and adults in the communities were extracted from the 2020 village census register obtained from the NTDs control department in the LGA. A total of 6,012 persons comprising 2531 children and 3481 adults were enumerated. As a further step, a sample-size was determined using the formula; n_s=n/(1+(n/N) ) and n= (z^2 p(1-p))/d^2 , as described by Lwanga et al [18], where ns is the required sample size and N is the target population. We assumed a prevalence (p) of 50% since there are no previous baseline data on schistosomiasis in the five communities, a relative precision (d) of 5% and a confidence level of 95% which corresponds to a z score of 1.96. The minimum sample size determined, therefore, was 362 i.e., an average of 72 persons per community. However, the recruitment of participants extended beyond the estimated sample size, considering the aim of identifying factors associated with the outbreak.

For the selection of study participants, we employed a total sampling methodology, following the method previously described by [19]. Community sensitization and advocacy visits were made to the community leaders and other stakeholders. This was followed by mobilization of eligible community members using town announcers to participate in the study. Only residents of the community, who are above age of 5 years, can provide consent or assent, and has lived within the community in the last 3 months were recruited into the study. Enrollment of participants and collection of samples took place at a central point in the community, provided by the community leader. This point has a secluded space for administering study questionnaires and sorting of samples before transporting them to the laboratory. The number of consenting participants varied across the communities, hence giving an unequal number of persons recruited. 

C4: Comprehensive description of the simple logistic regression procedure in the data management and data analysis section.

R4: Thank you very much editor. We have reworked this section to provide more comprehensive description of the logistic regression.

Data management and analysis 

Data obtained were downloaded from the remote server by the biostatistician, and imported into Microsoft Excel for sorting before analysis in SPSS 20.0 software. Data on socio-demographic characteristics and water contact behavior were considered as independent variable, while prevalence of infection was considered as dependent variable. Data were first subjected to descriptive statistics including frequencies and cross-tabulations, then followed by Pearson chi-square statistics to test for associations between the variables. Variables that were associated with infection were considered significant only when P <0.05. Subsequently, variables were subjected to univariate analyses i.e., logistic regression, to estimate the magnitude of association between infection data and other variables. Potential risk factors were entered into the model as covariates using bidirectional stepwise entry method. Reference category was formulated for categorical variables before analysis and observations with missing values for any variable were excluded from the analysis. Predictive index in the model is represented as Exp(𝐵) which is the odd-ratio (OR). A 95% confidence interval (CI) was calculated for the OR, and values were considered statistically significant when the CI does not include 1 and the P < 0.05.

C5: To add 95% confidence intervals for prevalence ratios in Table 2.

R5: Thank you very much for this comment, which has improved our manuscript. The 95% confidence intervals have been included in Table 2. We are very grateful for the comment.

C6: Both unadjusted and adjusted odds should be included in Tables 3 and 4.

R6: We limited our analyses to univariate analyses, because the variables were not significant in the multivariate model. Thank you very much for your comments. 

Reviewer #1: 

C1: The manuscript is well written. The edits are very few such as italicizing Schistosoma and just minimal grammatical lapses and better sentence construction. My only major question is the sample size, how it was computed and what proportion of the computed sample size is represented by the number of actual respondents.

R1: Thank you very much for your very valuable comment, which has improved the quality of our manuscript. We have worked through the manuscript and corrected the grammatical errors, and where possible, we constructed several sentences. Please find them as highlight in the tracked version of the manuscript.

Also, we have now provided very explicit detail on how the sample-size were estimated and how the participants were recruited.

Please find below

Sample size determination and selection of study participants

As an initial step, the total population of school-aged children and adults in the communities were extracted from the 2020 village census register obtained from the NTDs control department in the LGA. A total of 6,012 persons comprising 2531 children and 3481 adults were enumerated. As a further step, a sample-size was determined using the formula; n_s=n/(1+(n/N) ) and n= (z^2 p(1-p))/d^2 , as described by Lwanga et al [18], where ns is the required sample size and N is the target population. We assumed a prevalence (p) of 50% since there are no previous baseline data on schistosomiasis in the five communities, a relative precision (d) of 5% and a confidence level of 95% which corresponds to a z score of 1.96. The minimum sample size determined, therefore, was 362 i.e., an average of 72 persons per community. However, the recruitment of participants extended beyond the estimated sample size, considering the aim of identifying factors associated with the outbreak.

For the selection of study participants, we employed a total sampling methodology, following the method previously described by [19]. Community sensitization and advocacy visits were made to the community leaders and other stakeholders. This was followed by mobilization of eligible community members using town announcers to participate in the study. Only residents of the community, who are above age of 5 years, can provide consent or assent, and has lived within the community in the last 3 months were recruited into the study. Enrollment of participants and collection of samples took place at a central point in the community, provided by the community leader. This point has a secluded space for administering study questionnaires and sorting of samples before transporting them to the laboratory. The number of consenting participants varied across the communities, hence giving an unequal number of persons recruited. 

REVIEWER’S REPORT.

Summary

The paper is addressing a key issue in public health i.e., An outbreak of schistosomiasis in Takum LGA in Northeastern Nigeria. However, the paper has some shortfalls and I suggest you consider the following:

C1: The methodology needs some clarification, administration of drugs to people was larger than the sample surveyed. The use of error bar seems misleading and needed to be re-address

R1: Thank you very much for the valuable comments. Foremost, we explained within the manuscript that the epidemiological team responded to the outbreak by implementing mass administration of praziquantel to all eligible persons in the communities through the usual door-door preventive chemotherapy approach as recommended by the WHO. This approach doesn’t require prior diagnosis of infection before administration of tablets. Prior to the COVID-19 pandemic, this treatment is planned biennially (twice in a year) by the LGA NTD unit, but was ceased due to the COVID-19 pandemic. However, because of the outbreak reported, it became imperative to resume treatment in this area, which coincided with the time the epidemiological investigation was conducted. For this reason, the number of persons treated were larger in number than those surveyed in the study. 

Secondly, for the figures, we used standard deviation to illustrate the margin between the individual infection data. The authors prefer to use this metric in place of standard error which measures the accuracy of estimations.

C2: Though the paper was well presented, however, some aspect needed for follow a chronological order so as to have free flow when reading the paper if accepted for publication.

R2: Thank you very much for this valuable comment. We have revised this section to allow more thoughtful flow. We really appreciate the reviewer for pointing this out. We have revised several parts of the manuscripts to allow more chronological flow.

C3: The gap which was identified needed to be filled through right discussion on good implementation research

R3: Under the recommendation paragraphs and conclusion remarks, we have re-emphasized how to address the gaps that was identified. This comment was also addressed under the more specific comments raised by the reviewer (C25)

C4: The discussion needs to be aligned to the key results of the study and avoid reference to stuff that was note presented in the results.

R4: Thank you for this comment. We really appreciate your inputs. We have also addressed this comment under the more specific comments raised by the reviewer (C22 and C24)

C5: The present manuscript requires editing for grammar, spelling errors and further concision.

R5: We take this opportunity to appreciate the reviewer for the invaluable comments. We have worked through the manuscript and have corrected the grammar, spelling errors and deleted some unnecessary sentences 

Other comments

C1: Line 2. Abstract, kindly recast, there cannot be an outbreak of haematuria, it is just one of the symptoms of schistosomiasis

R1: Thank you very much for this valuable comment. We think PloS ONE goes with the American style.

C2; Page 2, keyword on hematuria, Please, be consistent with the choice of your English, it's either you go with American style of British.

R2: Thank you very much for this comment. 

C3: Page 10, Line 3 under Introduction, "one of the major and most.." please, kindly recast, it is seems ambiguous!

R3: Thank you very much for this comment. We have revised it to “one of the most” 

C4; Page 11, Line 1, For haematuria as a sign of the infection, most times, visible blood is noticed, you know?

R4: Thank you very much for this comment. We have revised the word symptom to sign. We thank you for this valuable addition.

C5: Page 22, Paragraph 2, Line 1, what is the unit of your age here?

R5: Thank you very much for this comment. We have revised it to “Children under 15 years of age….

C6: Page 22, Paragraph 2, Line 8 "LGA" is appearing for the first here, yet, you had only "government areas" here.?????

R6: Thank you very much for this comment. We have revised it to “local government areas….

C7: Page 22, Paragraph 2, Line 13, I would suggest, you allow your information to flow in a chronological order.

R7: Thank you very much for this valuable comment. We have revised it to allow more thoughtful flow. Please see the revised text below:

Nigeria is one of the schistosomiasis endemic countries in Africa [1], with 36 states and 774 local government areas (LGAs). About 708 LGAs had been mapped by the Federal Ministry of Health (FMoH), with 608 of them being endemic [16]. Since 2009, treatment with praziquantel commenced in 27 states with the support of WHO, UNICEF and partner organizations such as Mission to save the helpless (MITOSATH), Sightsavers, AMEN foundation among others [16]. Taraba, was among the states in mapped for schistosomiasis in 2010 and subsequently in 2014 [16,17]. The state is located in the northeastern region of the country, and has 16 LGAs.

C8: Page 12, Line 3, how many Takum community/LGA do you have in that LGA???

R8: Thank you very much for this valuable comment. We have highlighted this and revised accordingly. We have 2 different communities with the name Takum A and Takum B. Thanks for pointing this out.

C9: : Page 12, Line 7, ????

R9: Thank you very much for this valuable comment. We have highlighted this and revised accordingly. We have replaced the word hematuria with schistosomiasis

C10: Page 12, Line 11, Kindly correct your error here!!!!!

R10: Thank you very much for pointing out this error. We have deleted the repeated word.

C11: Page 13, under study area, Are you the first to carry out an investigation in that LGA? If your answer is NO. kindly, provide the reference for the coordinate provided here!!!!

R11: Thank you very much for this. We have provided a reference to support the description of the study area.

C12: Page 13, Line 4, under study design, kindly recast, this sentence is ambiguous!!!!!!!

R12: Thank you very much for pointing this out. We have reworked the sentence to provide more clarity

Please see below

The five study communities (Barkin lissa, Birama, Gamga, Shibong and Takpa) were randomly selected using the paper ballot approach out of the eleven communities.

C13; Page 13, Line 9, under study design, Again, you sentences should be in chronological order. I didn't find these statement easy to read.

R13: Thank you very much for pointing this out. We have reworked the sentence to provide more clarity

Please see below

Preliminary advocacy visits were made to the NTD control unit closer to the selected communities (i.e., ward level), prior to the epidemiological investigation

C14; Page 13, Line 3, under sample size, You invited all members of the community for the study, yet, you stated before that you had a method of choosing who to participate????? OR 

does the study involved a purposive method????

R14: Thank you very much for your very valuable comment, which has improved the quality of our manuscript. We have now provided very explicit details on how the sample-size were estimated and how the participants were recruited.

Please find below

Sample size determination and selection of study participants

As an initial step, the total population of school-aged children and adults in the communities were extracted from the 2020 village census register obtained from the NTDs control department in the LGA. A total of 6,012 persons comprising 2531 children and 3481 adults were enumerated. As a further step, a sample-size was determined using the formula; n_s=n/(1+(n/N) ) and n= (z^2 p(1-p))/d^2 , as described by Lwanga et al [18], where ns is the required sample size and N is the target population. We assumed a prevalence (p) of 50% since there are no previous baseline data on schistosomiasis in the five communities, a relative precision (d) of 5% and a confidence level of 95% which corresponds to z score 1.96. The minimum sample size determined, therefore, was 362 i.e., an average of 72 persons per community. However, the recruitment of participants extended beyond the estimated sample size, considering the aim of identifying factors associated with the outbreak.

For the selection of study participants, we employed a total sampling methodology, following the method previously described by [19]. Community sensitization and advocacy visits were made to the community leaders and other stakeholders. This was followed by mobilization of eligible community members to participate in the study using town announcers. Only residents of the community, who are above age of 5 years, can provide consent or assent, and has lived within the community in the last 3 months were recruited into the study. Enrollment of participants and collection of samples took place at a central point in the community, provided by the community leader. This point has a secluded space for administering study questionnaires and sorting of samples before transporting them to the laboratory. The number of consenting participants varied across the communities, hence giving roughly an unequal number of persons recruited. 

C15: Page 16, Line 5, Giving of these incentives looks as if you used those items to entice them for the study which it seems not ethical, if you had provided this information to your ethical review committee, maybe, it may have been spotted and provide necessary advice!!!!!!!!

R15: Thank you very much for your comments. These incentives were not used to entice the participants, as they were given even to those that refused to participate in the study but visited the collection point. This information was provided in our letter of request, it was highlighted that the immediate benefits from the study would be provision of treatment to all eligible members of the communities, and incentives such as bar soaps to reinforce positive hygiene behaviors and bisquits to stimulate community integration especially during a tensed situation which the pandemic has presented. As such we guided the time of introduction of the incentives, ensuring it is non-coercive and doesn’t influence their participation in the research 

Thank you very much for this valuable comment, and we have added some lines to support the sentence.

Please see below

All eligible and non-eligible participants were treated with 400mg/kg of praziquantel as an immediate benefit of the research investigation. In addition to this, incentives such as bar soaps and bisquits were given to reinforce positive hygiene behaviors and stimulate community integration during a tensed situation which the pandemic has presented. These incentives were provided in a non-coercive manner, ensuring they do not influence the decision of the community members to participate or decline participation in the research 

C16: Page 16, Line 12, under parasitological assessment, Did you divide the infection into categories????

R16: Yes the infection were divided into categories. One category for S. haematobium and the other for S. mansoni. Thank you for your comment.

C17: Page 16, Line 2, under treatment, The use of your "all" is confusing. do you mean all infected individual or all communities members who came for the test?

R17: Thank you very much for your comment. Yes we treated all eligible members of the community, irrespective of their participation in the study. This followed the routine guidelines for implementing MDA in the country. Door to door visitation was made and praziquantel was administered to every member of the community.

We have improved our text to highlight this.

C18: Page 18, under results, Line 4, What is the basis for checking for the significant difference here????

R18: Thank you for your comments here. Really there is no major basis here other than the fact that it gives prior information that the distribution of participants varied significantly by gender across the communities. Which is quite typical of most cross-sectional epidemiological studies. But we don’t think it add much value to the paper, or reduces it. But we would love to retain them if the reviewers don’t mind.

C19: Page 20, title,????

R19: Thank you very much for your comment. We really appreciate it

C20: Page 20, Line 1, under treatment data, 432 participated in the investigation, you treated 3,580 people. How do you explain the rationale for this action? Did you just treat people without conducting an appropriate test?

R20: Thank you very much for the valuable comments. Foremost, we explained within the manuscript that the epidemiological team responded to the outbreak by implementing mass administration of praziquantel to all eligible persons in the communities through the usual door-door preventive chemotherapy approach as recommended by the WHO. This approach doesn’t require prior diagnosis of infection before administration of tablets. Prior to the COVID-19 pandemic, this treatment is planned annually (once in a year) by the LGA NTD unit, but was ceased due to the COVID-19 pandemic. However, because of the outbreak reported, it became imperative to resume treatment in this area, which coincided with the time the epidemiological investigation was conducted. For this reason, the number of persons treated were larger in number than those surveyed in the study. 

C21: Page 20, line 3, under treatment data, Please what was the rationale for this????

R21: Thank you very much for your comments. The response to this comment has been provided above. We really appreciate your contributions.

C22: Page 21, Line 2, under discussion, This statement is not convincing enough!!!!

R22: Thank you very much for pointing this to us. This really improved the flow of our text. We have modified the text, and added a reference to support our statement here.

Please see below

Since 2014, Takum LGA has benefitted from three biennial rounds of MDA, which targeted 75% of its school-aged children. In these years, the therapeutic coverage were optimal, surpassing the 75% national targets, as such, the outbreak of urogenital schistosomiasis was unexpected [17].

C23: Page 21, Paragraph 2, line 1, ????????

R23: Thank you very much, we have improved the text, and changed the word from hematuria to schistosomiasis.

C24: Page 21, Paragraph 2, Line 4, You are discussing what you did not show in your result, why???

R24: Thank you very much for this observation. We have provided the results been presented here on Table 2. Now with your comments we found it necessary to improve the table and also include a footnote with thte table to describe the categories of endemicity based on prevalence value. We take this time to thank you once more for the valuable comments.

Please see below

Categories of Endemicity: 

aLow endemicity when prevalence is between 1-9.9%;

bModerate endemicity when prevalence is between 10-49.9% 

cHigh endemicity when prevalence is above 50%)

C25: Page 23, Line 1, under conclusion, since you identified a gap in the implementation of drug delivery by FMoH and other for the treatment of the disease, I thought you will focus more on how to address right implementation of drug delivery but little or nothing is said about it in this conclusion and recommendation.

R25: Thank you very much once again for this very valuable comment. We have now adjusted the recommendation and conclusion lines, to buttress these points. Thanks.

Please see below

Under recommendations:

It is therefore imperative to consider; (1) investments in effort targeted at reclassifying the LGA as highly endemic, and adjusting the MDA thresholds from the biennial cycle to an annual cycle

Under conclusions:

Until now, Takum was classified to be a low endemic LGA and had been receiving biennial treatment. This outbreak and our prevalence reports highlight the need to re-classify the LGA as highly endemic, and adjust the MDA thresholds from the biennial cycle to an annual cycle

C26: Figure 2: Your error bar looks misleading, if you use standard error data for your error bar, I don't think you will have this high error bar for your data!!!!

R26: Thank you so much for all your wonderful comments, which were really helpful. For the figures, we used standard deviation to illustrate the margin between the individual infection data. The authors prefer to use this metric in place of standard error which measures the accuracy of estimations.

---

## [Decision Letter · Decision Letter 1]

26 Dec 2021

PONE-D-21-30831R1Schistosomiasis outbreak during COVID-19 pandemic in Takum, Northwest Nigeria: Analysis of Infection status and associated risk factorsPLOS ONE

Dear Dr. Mogaji,

Thank you for submitting your manuscript to PLOS ONE. After careful consideration, we feel that it has merit but does not fully meet PLOS ONE’s publication criteria as it currently stands. 

We look forward to receiving your revised manuscript.

Kind regards,

Khin Thet Wai, MBBS, MPH, MA

Academic Editor

PLOS ONE

Journal Requirements:

Reviewers' comments:

Reviewer's Responses to Questions

**Comments to the Author**

1. If the authors have adequately addressed your comments raised in a previous round of review and you feel that this manuscript is now acceptable for publication, you may indicate that here to bypass the “Comments to the Author” section, enter your conflict of interest statement in the “Confidential to Editor” section, and submit your "Accept" recommendation.

Reviewer #2: (No Response)

2. Is the manuscript technically sound, and do the data support the conclusions?

Reviewer #2: Yes

3. Has the statistical analysis been performed appropriately and rigorously? 

Reviewer #2: Yes

4. Have the authors made all data underlying the findings in their manuscript fully available?

Reviewer #2: Yes

5. Is the manuscript presented in an intelligible fashion and written in standard English?

Reviewer #2: Yes

6. Review Comments to the Author

Reviewer #2: The did a good job by updating tge manuscript, I really commend them. However, I will encourage the authors to look into the error bars in fifure 3 and 4, the error bars are too high giving the impression that there are so meny errors assosiated with the data collected. Thank you.

7. PLOS authors have the option to publish the peer review history of their article (what does this mean?). If published, this will include your full peer review and any attached files.

Reviewer #2: No

---

## [Author Response · Author response to Decision Letter 1]

27 Dec 2021

C1: They did a good job by updating the manuscript, I really commend them. However, I will encourage the authors to look into the error bars in figure 3 and 4, the error bars are too high giving the impression that there are so meny errors assosiated with the data collected. Thank you.

R1: Thank you very much for your very valuable comment, which has improved the quality of our manuscript. We agree completely with you. We have re-worked the error-bars using percentage error bars, because both SE and SD error bars were too high after trying them. We have provided new figures showing these new error bars. We take this opportunity to appreciate the reviewers and editor for the great efforts on our manuscript.

---

## [Editor Report · Decision Letter 2]

28 Dec 2021

Schistosomiasis outbreak during COVID-19 pandemic in Takum, Northwest Nigeria: Analysis of Infection status and associated risk factors

PONE-D-21-30831R2

Dear Dr. Mogaji,

We’re pleased to inform you that your manuscript has been judged scientifically suitable for publication and will be formally accepted for publication once it meets all outstanding technical requirements.

Kind regards,

Khin Thet Wai, MBBS, MPH, MA

Academic Editor

PLOS ONE
---

## [Editor Report · Acceptance letter]

12 Jan 2022

PONE-D-21-30831R2 

Schistosomiasis outbreak during COVID-19 pandemic in Takum, Northeast Nigeria: Analysis of Infection status and associated risk factors 

Dear Dr. Mogaji:

I'm pleased to inform you that your manuscript has been deemed suitable for publication in PLOS ONE. Congratulations! Your manuscript is now with our production department. 

Kind regards, 

on behalf of

Dr. Khin Thet Wai 

Academic Editor

PLOS ONE